# Mac-2 binding protein glycosylation isomer, the FIB-4 index, and a combination of the two as predictors of non-alcoholic steatohepatitis

**Ito Kiyoaki**[ORCID]**\*, Yoshio Sumida, Yukiomi Nakade, Akinori Okumura, Sayaka Nishimura, Mayu Ibusuki, Rena Kitano, Kazumasa Sakamoto, Satoshi Kimoto, Tadahisa Inoue, Yuji Kobayashi, Yoshitaka Fukuzawa, Masashi Yoneda**

Department of Gastroenterology, Aichi Medical University, Nagakute, Japan

\* kito@aichi-med-u.ac.jp

**Data Availability Statement:** All relevant data are within the manuscript.

## Abstract

Approximately 10% non-alcoholic fatty liver disease (NAFLD) cases progress to non-alcoholic steatohepatitis (NASH). Liver biopsy, the gold standard for diagnosing NASH and associated liver fibrosis, is invasive with a risk of life-threatening complications. Therefore, reliable non-invasive biomarkers for predicting NASH are required to prevent unnecessary liver biopsies. We evaluated the performance of two non-invasive fibrosis markers, Mac-2 binding protein glycosylation isomer (M2BPGi) and the FIB-4 index for predicting the fibrosis staging, NAFLD activity scoring (NAS) index, and NASH. We also analyzed the correlation between the two markers. The sensitivities, specificities, positive predictive values (PPV), and negative predictive values of the FIB-4 index, M2BPGi, and a combination of both markers for NASH diagnosis were evaluated. The M2BPGi and FIB-4 index showed a good performance in diagnosing NASH, the fibrosis stage, and the NAS index in NAFLD patients. While both markers were well-correlated with each other in most cases, no correlation was found in some patients. Compared with the FIB-4 index or the M2BPGi alone, a combination of the two showed a higher specificity, PPV, and accuracy for NASH diagnosis. The M2BPGi and the FIB-4 index are easily accessible and reliable liver fibrosis markers. Diseases other than liver disease may cause dissociation between the two markers, causing failure to predict NASH. However, the combination of both markers can compensate for their disadvantages. Because the PPV of the combination was relatively high, patients who test positive for both markers should undergo liver biopsy for NASH diagnosis.

## Introduction

Non-alcoholic fatty liver disease (NAFLD) is one of the most common causes of chronic liver disease worldwide [1]. With the increasing westernization of dietary intake, approximately 25% of the global adult population suffers from NAFLD [2]. In addition, NAFLD is considered an indicator of metabolic syndrome, a group of cardiovascular risk factors [3, 4]. Furthermore, non-alcoholic steatohepatitis (NASH), which is defined by steatosis, necroinflammation, and cytopathic changes in NAFLD, may progress to liver cirrhosis and hepatocellular carcinoma

**Funding:** The authors received no specific funding for this work.

**Competing interests:** Dr. Ito has received research funding from Gilead Sciences, Bristol-Myers Squibb, and AbbVie. Dr. Sumida has received honoraria from Mitsubishi Tanabe, Sumitomo Dainippon, AstraZeneca, Ono, and Taisho pharm and has received research funding from Bristol-Myers Squibb and Gilead Sciences. Dr.Yoneda has received research funding from AbbVie, Bayer Pharma, EA pharma, Otsuka Pharmaceutical, and Sumitomo Pharmaceutical. This does not alter our adherence to PLOS ONE policies on sharing data and materials.

(HCC) [5]. Approximately 20% of NASH cases can slowly progress to liver cirrhosis and HCC [6, 7]. While the pathogenesis of NASH remains unclear, many parallel hits from adipose tissue and the intestine are thought to promote liver inflammation [8]. Currently, liver biopsy is the gold standard for diagnosing NASH and associated liver fibrosis [6, 9]; however, liver biopsy is invasive and potentially life-threatening [10]. Therefore, non-invasive approaches are required to distinguish NASH from NAFLD to reduce unnecessary liver biopsies.

Mac-2 binding protein glycosylation isomer (M2BPGi) has been recently recognized as a novel serum marker for liver fibrosis by glycoproteomic biomarker screening studies [11, 12]. M2BPGi has been demonstrated to have multi-branching and sialylated *N*-glycans. In addition, M2BPGi is thought to recognize clustered LacNAc (Gal-GlcNAc) structures or GalNAc residues of *N*-glycans and *O*-glycans [11, 12]. We have previously published a meta-analysis to determine the predictive value of serum M2BPGi for liver fibrosis in chronic liver diseases such as chronic hepatitis B, chronic hepatitis C, alcoholism, NAFLD, NASH, autoimmune hepatitis, primary biliary cholangitis, and biliary atresia [13]; our findings indicated that this marker is useful in predicting chronic liver diseases with broad etiologies. In addition, M2BPGi is reportedly useful for NASH diagnosis [14]. However, since M2BPGi can be elevated due to causes other than liver diseases, there are limitations to using this marker alone to predict NASH.

The FIB-4 index was developed as a non-invasive panel for staging the liver of patients with hepatitis C virus (HCV) infection [15]. It is calculated from the patient's age, aspartate aminotransferase (AST) level, alanine aminotransferase (ALT) level, and platelet count measured in routine practice and are available to almost all patients with liver disease. It has recently been demonstrated that its performance characteristics for diagnosing advanced fibrosis in NAFLD cases are better than those of other similar panels (such as NAFLD fibrosis score, AST:ALT ratio, AST:platelet ratio index, etc.) that do not require additional testing [16]. The usefulness of the FIB-4 index in predicting advanced liver fibrosis has already been recognized [17]. However, because FIB-4 index can be elevated due to low platelet count rather than liver diseases, there are limitations to using this marker alone to predict NASH.

In this study, we performed liver biopsies to diagnose NASH in NAFLD patients. Furthermore, we have evaluated the diagnostic potential of M2BPGi and the FIB-4 index for liver fibrosis, NAFLD activity scoring (NAS), and NASH.

## Materials and methods

### Patients and diagnosis for NASH

This retrospective study recruited 93 biopsy-confirmed NAFLD patients treated at the Aichi Medical University Hospital between January 2012 and December 2018. Informed consent was obtained from all patients, and the study was approved by the ethics review board at the Aichi Medical University Hospital. NAFLD was diagnosed based on liver biopsy findings of steatosis in 5% or more of hepatocytes. The exclusion criteria included: 1) a history of other hepatic diseases, 2) a substance abuse-induced hepatic disorder, and 3) a history of alcohol abuse (>20 g of alcohol daily for women and >30 g of alcohol daily for men). All patients underwent a percutaneous needle biopsy using standard procedures. The collected specimens were embedded in paraffin blocks and stained with hematoxylin and eosin and Masson's trichrome stains. Then, two expert hepatologists blinded to the clinical data evaluated the specimens. An adequate liver sample was defined as being >1.5 cm in length and/or containing more than six portal tracts. Matteoni's classification was then used to confirm the presence of NASH [18]. Accordingly, NAFLD patients with ballooning hepatocytes (that is, Matteoni type 3) and those with liver fibrosis (that is, Matteoni type 4) were assigned to the NASH group

(n = 62), while patients in whom liver biopsy analysis revealed simple steatosis or steatosis with nonspecific inflammation were assigned to the NAFL group (n = 31). NAS was used to assess and quantify each sample [19], and the stages of steatosis (stages 0–3), lobular inflammation (stages 0–2), and hepatocellular ballooning (0–2) were quantified. Furthermore, individual fibrosis parameters were scored independently using the NASH Clinical Research Network scoring system [19].

## Clinical and biological data

The age and sex of the patients were recorded. Serum samples were collected immediately before or no more than 2 months after liver biopsy and were stored at –80˚C until analysis. These samples were analyzed to assess the concentrations of the following variables: AST, ALT, alkaline phosphatase (ALP), total bilirubin, albumin, cholinesterase, total cholesterol, platelet count, and prothrombin time. The FIB-4 index was calculated using Sterling et al.'s formula as follows: (age [years] × AST [IU/L]/(platelet count [×$10^9$/L] ×$\sqrt{}$ALT [IU/L]) [15].

## Measurement of M2BPGi

M2BPGi levels were quantified using an automated chemiluminescence enzyme immunoassay system (HISCL-800; Sysmex, Kobe, Japan) that was operated following the manufacturer's recommendations [12]. The measured values of M2BPGi conjugated to *Wisteria floribunda* agglutinin (WFA) were indexed with the values obtained using the following equation: cut-off index (COI) = ([M2BPGi]sample–[M2BPGi]NC)/([M2BPGi]PC–[M2BPGi]NC), where [M2BPGi] sample was the concentration of M2BPGi in the serum sample, NC was the negative control, and PC was a positive control. The PC was a calibration solution preliminarily standardized to yield a COI value of 1.0 [12].

## Statistical analyses

All statistical analyses were performed using STATA version 15.0 (Stata-Corp, College Station, TX, USA). Quantitative variables were expressed as means ± standard deviation (SD) unless specified otherwise. Categorical variables were compared using the chi-square test or the Fisher's exact test, as appropriate, and continuous variables were compared using the Mann-Whitney *U* test. $P < 0.05$ was considered statistically significant. Spearman's rank correlation coefficient were used when necessary. Receiver operating characteristic (ROC) curve analysis was performed to assess the diagnostic accuracies of the M2BPGi level and FIB-4 index for NASH based on the area under the curve (AUC) values. Diagnostic accuracy was expressed as diagnostic specificity (specificity), diagnostic sensitivity (sensitivity), positive predictive values (PPV), negative predictive values (NPV), and AUC.

## Results

### Baseline characteristics of the 93 patients with NAFLD at the time of liver biopsy

Table 1 shows the patients' characteristics at the time of liver biopsy. The mean age of the NASH group was significantly higher than that of the NAFL group ($P = 0.0001$). Furthermore, the percentage of women was significantly higher in the NASH group than in the NAFL group. The AST and ALT levels were significantly higher in the NASH group than in the NAFL group ($P = 0.0006$ and 0.046, respectively). The albumin and total cholesterol levels were significantly lower in the NASH group than in the NAFL group ($P = 0.003$ and $P = 0.04$, respectively). The platelet count and prothrombin time were significantly lower in the NASH

**Table 1. Baseline characteristics of the 93 patients with NAFLD at the time of liver biopsy.**

| Features | NASH (n = 62) | NAFL (n = 31) | P Value |
|---|---|---|---|
| Age (years) | 60.9±1.9 | 49.4±2.6 | 0.0001 |
| Female (%) | 35 (56.5%) | 10 (33.3%) | 0.028 |
| BMI (kg/m$^2$) | 26.8 ± 0.49 | 27.6 ± 0.88 | 0.417 |
| AST (IU/L) | 70.8 ± 40.7 | 41.9 ± 29.2 | 0.0006 |
| ALT (IU/L) | 91.1 ± 62.4 | 61.8 ± 72.4 | 0.046 |
| ALP (IU/L) | 274.9 ± 107.9 | 251.5 ± 63.4 | 0.269 |
| Total bilirubin (mg/dL) | 0.9 ± 0.4 | 0.8 ± 0.3 | 0.06 |
| Albumin (g/dL) | 4.2 ± 0.4 | 4.4 ± 0.3 | 0.003 |
| Total cholesterol (mg/dL) | 188.6 ± 30.3 | 210.9 ± 71.6 | 0.04 |
| LDL cholesterol (mg/dL) | 116.2 ± 28.9 | 120.3 ± 27.5 | 0.56 |
| Triglyceride (mg/dL) | 165.4 ± 86.4 | 229.3 ± 312.5 | 0.148 |
| Platelet (×10$^3$/μL) | 191.9 ± 71.7 | 241.1 ± 55.8 | 0.0012 |
| Prothrombin time (%) | 87.9 ± 18.2 | 102.5 ± 12.2 | 0.0006 |
| M2BPGi (COI) | 1.5 ± 0.9 | 0.6 ± 0.4 | < 0.0001 |
| FIB-4 index | 3.2 ± 2.5 | 1.3 ± 0.8 | 0.0001 |

NAFLD: non-alcoholic fatty liver disease; NASH: non-alcoholic steatohepatitis; NAFL: non-alcoholic fatty liver, BMI: body mass index; AST: aspartate aminotransferase; ALT: alanine aminotransferase; ALP: alkaline phosphatase; LDL: low-density lipoprotein; M2BPGi: Mac-2 binding protein glycosylation isomer; COI: cut-off index

group than in the NAFL group ($P$ = 0.0012 and 0.0006, respectively). Furthermore, the M2BPGi level ($P$ < 0.0001) and the FIB-4 index ($P$ = 0.0001) were significantly higher in the NASH group than in the NAFL group.

## Evaluation of the M2BPGi levels and FIB-4 indices for estimating the progression of liver fibrosis

The boxplots of the M2BPGi level and the FIB-4 index for fibrosis staging are shown in Fig 1A and 1B, respectively. While the median M2BPGi level at stage 4 was lower than that at stage 3, the M2BPGi values gradually increased with fibrosis progression (R = 0.553, $P$ < 0.0001). The FIB-4 index also gradually increased with fibrosis progression (R = 0.577, $P$ < 0.0001).

## Evaluation of the M2BPGi levels and the FIB-4 Index for estimating the progression of NAS

The scatter plots of the M2BPGi level and the FIB-4 index regarding the NAS are shown in Fig 2A and 2B, respectively. The M2BPGi level and FIB-4 index showed similar tendencies for estimating NAS progression. Both values gradually increased until the NAS score was 5; however, both values were lower for scores 6 and 7. Spearman's rank correlation coefficient between NAS and M2BPGi and between NAS and FIB-4 index was R = 0.332 ($P$ = 0.0012) and R = 0.333 ($P$ = 0.0011), respectively.

## Comparison of the AUCs of the M2BPGi level and the FIB-4 index for predicting NASH

The ROC curves of the M2BPGi and the FIB-4 index, computed to determine their predictive value for NASH, are shown in Fig 3. The AUC of M2BPGi (0.830) was superior to that of the

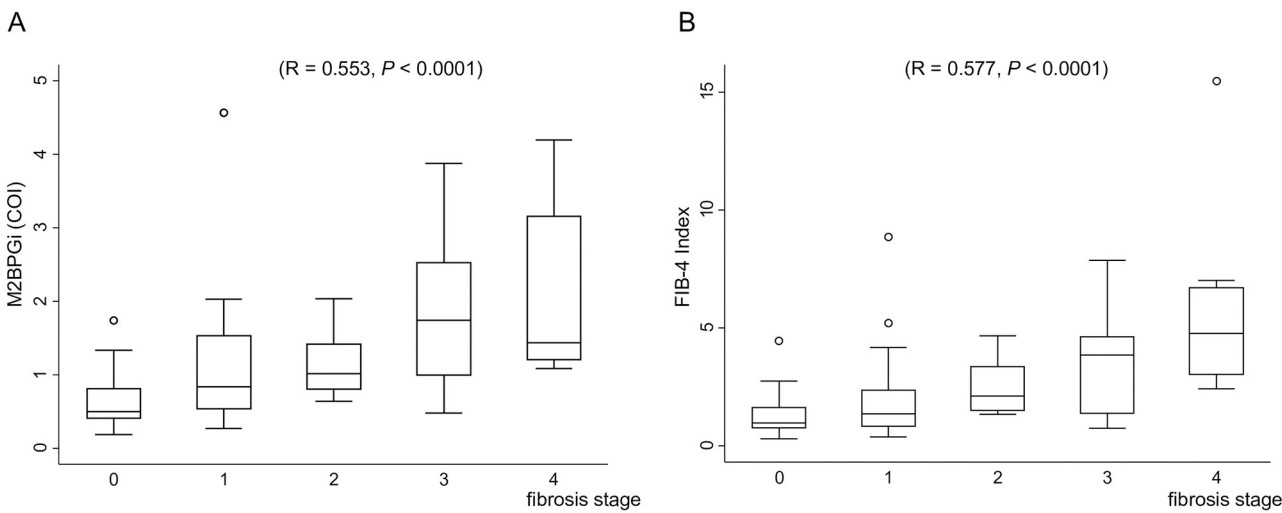

**Fig 1. Boxplot of (A) M2BPGi and (B) the FIB-4 index for fibrosis staging.** The box represents the interquartile range. The whiskers indicate the highest and the lowest values, while the dots represent outliers. The line across the box indicates the median value. M2BPGi: Mac-2 binding protein glycosylation isomer.

FIB-4 index (0.792). The best cut–off values for NASH diagnosis using M2BPGi and the FIB-4 index were 0.6 and 1.23, respectively.

## Diagnosing NASH by using M2BPGi, FIB-4 index, and a combination of both markers

The sensitivity and NPV of M2BPGi alone were superior to those of the FIB-4 index alone and the combination of the two markers. In addition, the specificity, PPV, and accuracy of the combination of the two markers were superior to those of M2BPGi and the FIB-4 index alone (Table 2).

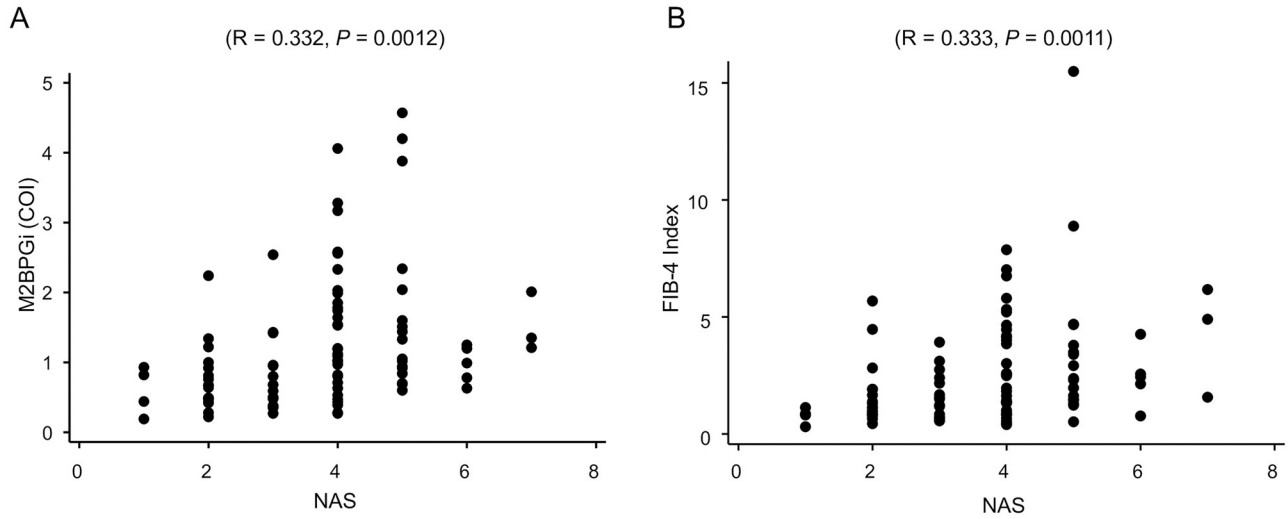

**Fig 2. Dot plots of (A) M2BPGi and (B) the FIB-4 index for the non–alcoholic fatty liver disease activity score.** M2BPGi: Mac-2 binding protein glycosylation isomer.

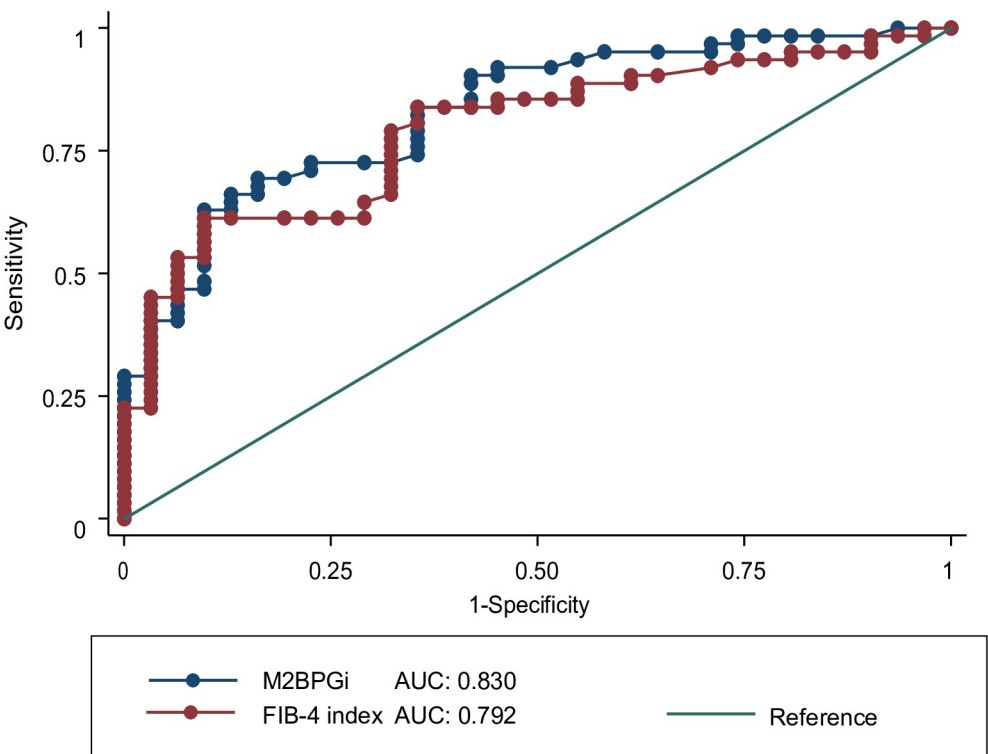

**Fig 3. Comparing the ROC curves for the M2BPGi and FIB-4 index performances in diagnosing non–alcoholic steatohepatitis.** M2BPGi: Mac-2 binding protein glycosylation isomer; ROC: receiver operating characteristic curve; AUC: area under the ROC curve.

## Relationship between M2BPGi and FIB-4 index

The M2BPGi level and the FIB-4 index significantly corelated (R = 0.722, $P < 0.0001$). The nine patients with a divergence between M2BPGi and the FIB-4 index were numbered as cases 1–9. In cases 1–5, the M2BPGi was relatively higher than the FIB-4 index. In contrast, the M2BPGi was relatively lower than the FIB-4 index in cases 6–9 (Fig 4).

## Characteristics of the patients with discrepancies between M2BPGi and the FIB-4 index

Table 3 shows the characteristics of the patients who exhibited divergence between the M2BPGi and the FIB-4 index in Fig 3. Cases 1–5 presented with non-liver-related diseases,

**Table 2. The cut-off values, sensitivity, specificity, PPVs, NPVs, and accuracy of the M2BPGi level, FIB-4 index, and a combination of both markers for diagnosis of NASH.**

|  | Cut off value | Sensitivity (%) | Specificity (%) | PPV (%) | NPV (%) | Accuracy |
|---|---|---|---|---|---|---|
| **M2BPGi** | 0.6 | **90.3** | 58.1 | 81.2 | **75.0** | 79.6 |
| **FIB-4 index** | 1.23 | 79.0 | 67.7 | 83.1 | 61.8 | 75.3 |
| **Combination** | 0.6 [a] and 1.23 [b] | 82.3 | **77.4** | **87.9** | 68.6 | **80.6** |

Bold type indicates the highest value among the three groups.

PPV: positive predictive value; NPV: negative predictive value; M2BPGi: Mac-2 binding protein glycosylation isomer; NASH: non-alcoholic steatohepatitis

[a]: for M2BPGi;

[b]: for FIB-4 index

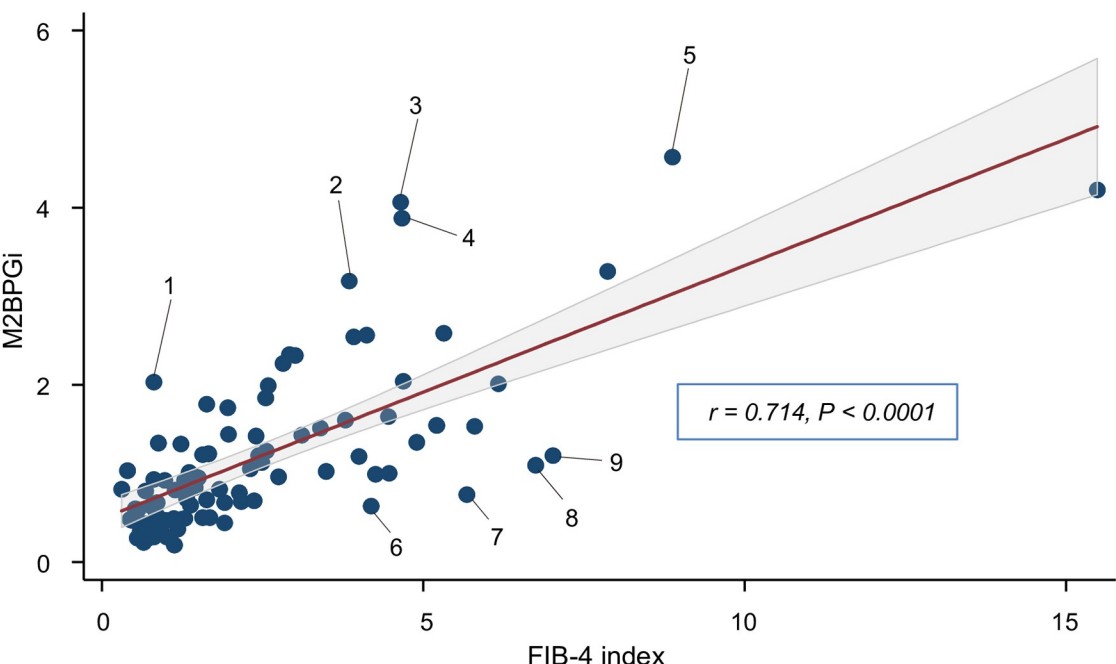

**Fig 4. Scatter plot shows the association between M2BPGi and the FIB-4 index.** The Spearman's rank correlation coefficient (r) is 0.722 (P < 0.0001). M2BPGi: Mac-2 binding protein glycosylation isomer.

such as diabetes mellitus, rheumatoid arthritis, hypertension, breast cancer, and hyperlipidemia. In cases 6 and 8, the platelet counts were relatively low despite normal values of albumin, total bilirubin, and prothrombin time. The AST levels in cases 7 and 9 were higher than those in the other cases (Table 3).

**Table 3. Characteristics of the patients exhibiting discrepancies between the M2BPGi level and the FIB-4 index.**

| Case | Age (years) | Gender | BMI (kg/m²) | Platelet (×10³/μL) | AST (IU/L) | Albumin (g/dL) | T–bil (mg/dL) | T–cho (mg/dL) | PT (%) | FIB-4 | M2BPGi (COI) | Diagnosis | Brunt (grading) | Brunt (staging) | NAS | Non-liver related diseases |
|---|---|---|---|---|---|---|---|---|---|---|---|---|---|---|---|---|
| 1 | 45 | F | 22.1 | 32.2 | 96 | 3.8 | 0.8 | 239 | 111 | 0.81 | 2.03 | NASH | 1 | 1 | 4 | Leukemia (after treatment) |
| 2 | 79 | F | 27.3 | 19.5 | 57 | 3.3 | 0.9 | 133 | 68 | 3.85 | 3.17 | NASH | 2 | 4 | 4 | DM, Rheumatoid arthritis |
| 3 | 78 | F | 25.3 | 10.2 | 36 | 4.3 | 1.1 | 166 | 60 | 4.65 | 4.06 | NASH | 2 | 4 | 4 | Hypertension |
| 4 | 67 | F | 30.4 | 8.7 | 50 | 4.1 | 0.9 | 145 | 99 | 4.67 | 3.88 | NASH | 3 | 3 | 5 | Breast cancer |
| 5 | 66 | F | 26.4 | 5.5 | 37 | 4.1 | 2.0 | 145 | 54 | 8.88 | 4.57 | NASH | 1 | 1 | 5 | Hyperlipidemia, DM |
| 6 | 65 | M | 31.6 | 11.4 | 54 | 4.2 | 1.3 | 192 | 94 | 4.19 | 0.63 | NASH | 1 | 2 | 4 | (-) |
| 7 | 69 | M | 27.0 | 19.5 | 230 | 3.3 | 2.0 | 160 | 51 | 5.68 | 0.76 | NASH | 1 | 3 | 2 | (-) |
| 8 | 68 | F | 24.9 | 9.4 | 64 | 4.2 | 1.1 | 177 | 72 | 6.75 | 1.09 | NASH | 2 | 4 | 4 | (-) |
| 9 | 84 | F | 23.1 | 14.3 | 158 | 3.9 | 1.0 | 192 | 83 | 7.02 | 1.2 | NASH | 1 | 4 | 4 | (-) |

BMI: body mass index; AST: aspartate aminotransferase; T-bil: total bilirubin; T-cho: total cholesterol, PT: prothrombin time; M2BPGi: Mac-2 binding protein glycosylation isomer; COI: cut-off index; NASH: non-alcoholic steatohepatitis; NAS: non-alcoholic fatty liver disease activity score; DM: diabetes mellitus

## Discussion

This study shows that the M2BPGi level and the FIB-4 index are independent diagnostic markers of liver fibrosis, NAS, and NASH. M2BPGi and FIB-4 index gradually increased with fibrosis progression. There was an overall significant difference in the association between the NAS score and the M2BPGi or FIB-4 index. While there was no significant difference between scores 6 and 7 and other scores, scores 6 and 7 tended to be low. Therefore, because steatosis and lobular inflammation decrease when fibrosis is severely progressed, fibrosis is considered more advanced in NAS scores 4 and 5 than in NAS scores of 6 and 7. In this study, both markers mostly showed a good correlation with each other; however, this was not the case in some patients, and one had a weaker predictive value for NASH diagnosis. A combination of both markers had a better predictive value for NASH than each maker alone because both compensated foreach other's disabilities.

Because M2BPGi was developed using the sera derived from patients with HCV [12] and is influenced by cytokine responses and the viral protein associated with HCV infection [20], the titer of M2BPGi was relatively higher among patients with HCV infection than among patients with other pathologies [13]. We have also reported in a previously published meta-analysis that the diagnostic COI of M2BPGi for liver fibrosis differs among various pathologies (including HBV, HCV, primary biliary cholangitis, NAFLD, and NASH) [13]. It has been previously reported that M2BPGi values help assess the stages of liver fibrosis [21] and predict HCC development [22] in patients with NAFLD. While the COI was relatively low (0.6) in this study, NASH could be predicted using M2BPGi levels. M2BPGi showed higher sensitivity and NPV for NASH diagnosis than the FIB-4 index. Furthermore, the combination of both markers had a better specificity, PPV, and accuracy for NASH diagnosis than the M2BPGi level or the FIB-4 index alone. Therefore, if patients with NAFLD present with M2BPGi level >0.6 and FIB-4 index >1.23, a liver biopsy must be performed for NASH diagnosis.

Inaccurate prediction of NASH using M2BPGi alone may be due to a higher titer of M2BPGi caused by diseases other than liver diseases. It has been reported that M2BPGi was upregulated in patients with idiopathic pulmonary fibrosis [23], chronic pancreatitis [24], or atherosclerosis [25]. Shirabe et al. suggested that the M2BPGi level may be associated with cell adhesion, growth regulation, cytokine production, and T cell apoptosis [26]. Kianoush et al. also suggested an association between M2BPGi levels and the M2 polarization of macrophages [27]. Therefore, a higher activity of macrophages in pulmonary fibrosis, chronic pancreatitis, or atherosclerosis may be due to a higher M2BPGi titer and an inaccurate prediction of NASH. In this study, cases 1–5 had M2BPGi titers that were higher than the corresponding FIB-4 indices, and these patients presented with non-liver-related diseases, such as leukemia, diabetes mellitus, rheumatoid arthritis, hypertension, breast cancer, and hyperlipidemia. These diseases may be associated with macrophage activation and may cause higher M2BPGi titers even without liver fibrosis.

The FIB-4 index was developed as a non-invasive marker for predicting liver fibrosis in patients with an HIV and HCV co-infection [15]. It was recently demonstrated that the performance of the FIB-4 index in predicting advanced fibrosis in NAFLD cases was better than that of other non-invasive markers, such as the AST/ALT ratio, cirrhosis determinant score, AST/platelet ratio, Goteburg university cirrhosis index, AST/platelet ratio index, BARD score, and NAFLD fibrosis score [16]. A previous study showed that a FIB-4 index $\geq$ 2.67 and a FIB-4 index $\leq$ 1.30 had a higher PPV and NPV, respectively, for advanced fibrosis (stages 3–4) in patients with NAFLD [16]. The FIB-4 index was calculated as: (age [years] × AST [IU/L]/platelet count [$\times 10^9$/L] $\times \sqrt{}$ALT [IU/L]) [15]. Therefore, the index would be higher in the patients with a lower platelet count due to diseases other than liver fibrosis (including blood disorders such as idiopathic thrombocytopenic purpura or thrombotic thrombocytopenic purpura [28]

and viral infection such as influenza [29]). In cases 6 and 8, the platelet counts were relatively low despite normal values of albumin, total bilirubin, and prothrombin time. Higher AST levels also cause an overestimation of the FIB-4. AST levels were higher in cases 7 and 9 than in other cases. When both markers are combined, M2BPGi can compensate for the predictive failure of the FIB-4 index in cases with lower platelet counts and higher AST levels due to diseases other than liver fibrosis.

The estimated global prevalence of NAFLD is 20–30%, and 67–75% in the general and obese populations, respectively [30, 31]. In Asia, the prevalence of NAFLD in the general population is 15–30% [32, 33]; the incidence of NASH is 1–3% in the adult Japanese population and approximately 6% in the Western population [34, 35]. Liver biopsy remains the gold standard for NASH diagnosis and staging. Therapeutic trials for NASH also require a liver biopsy to establish an initial diagnosis of NASH and document the treatment response [36]. However, liver biopsy is an invasive procedure and is limited by sampling error, high cost, procedure–related complications, and observer variability, even when performed by expert pathologists [37, 38]. Therefore, it is important to reduce unnecessary liver biopsies. There is an unmet need for robust, reliable, cost-effective, and non-invasive biomarkers that will allow practitioners to diagnose and stage NAFLD and also monitor NAFLD progression. Such biomarkers could also be a valuable addition to the current design of clinical trials on NASH [39].

This study has some limitations. It is a small-scale, hospital-based retrospective cohort study; therefore, it has inherent selection bias. Because most NAFLD cases in actual practice involve NAFL, we must carefully adapt our results to a population-based cohort. Therefore, further prospective studies are needed.

In conclusion, both the M2BPGi and the FIB-4 index are easily accessible and reliable liver fibrosis markers. Because each marker alone has some weaknesses for predicting NASH, a combination of both compensates for their individual weaknesses. Furthermore, because the PPV of the combination of M2BPGi and the FIB-4 index for NASH was high, patients deemed "positive" by both markers should undergo liver biopsy for NASH diagnosis. When M2BPGi is high and the FIB-4 index is low, the presence of diabetes mellitus, rheumatic disease, arteriosclerosis, and malignancy should be examined. Furthermore, when the M2BPGi is low and the FIB-4 index is high, other causes of platelet decrease or AST increase other than liver fibrosis should be examined. If no cause other than liver fibrosis is found to cause the discrepancy between M2BPGi and FIB-4 index, liver biopsy should be considered again.

## Author Contributions

**Conceptualization:** Ito Kiyoaki.

**Data curation:** Yoshio Sumida, Yukiomi Nakade, Akinori Okumura, Sayaka Nishimura, Mayu Ibusuki, Rena Kitano, Kazumasa Sakamoto, Satoshi Kimoto, Tadahisa Inoue, Yuji Kobayashi, Yoshitaka Fukuzawa, Masashi Yoneda.

**Formal analysis:** Ito Kiyoaki.

**Investigation:** Ito Kiyoaki, Yoshio Sumida.

**Methodology:** Ito Kiyoaki, Yoshio Sumida.

**Project administration:** Ito Kiyoaki.

**Supervision:** Ito Kiyoaki.

**Writing – original draft:** Ito Kiyoaki.

**Writing – review & editing:** Ito Kiyoaki.

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
