## [Decision Letter · Decision Letter 0]

13 Sep 2022

PONE-D-22-23754Mac-2 binding protein glycosylation isomer, the FIB-4 index, and a combination of the two as predictors of non-alcoholic steatohepatitisPLOS ONE

Dear Dr. Ito Kiyoaki,

Thank you for submitting your manuscript to PLOS ONE. After careful consideration, we feel that it has merit but does not fully meet PLOS ONE’s publication criteria as it currently stands. Therefore, we invite you to submit a revised version of the manuscript that addresses the points raised during the review process.  The study has meri and we would like to encourage resubmission.  Please submit your revised manuscript within Oct 28 2022 11:59PM. If you will need more time than this to complete your revisions, please reply to this message or contact the journal office at plosone@plos.org. Please include the following items when submitting your revised manuscript:A rebuttal letter that responds to each point raised by the academic editor and reviewer(s). You should upload this letter as a separate file labeled 'Response to Reviewers'.A marked-up copy of your manuscript that highlights changes made to the original version. You should upload this as a separate file labeled 'Revised Manuscript with Track Changes'.An unmarked version of your revised paper without tracked changes. You should upload this as a separate file labeled 'Manuscript'.

We look forward to receiving your revised manuscript.

Kind regards,

Gianfranco D. Alpini

Academic Editor

PLOS ONE

Journal Requirements:

"I have read the journal's policy and the authors of this manuscript have the following competing interests:  Dr. Ito has received research funding from Gilead Sciences, Bristol-Myers Squibb, and AbbVie. Dr. Sumida has received honoraria from Mitsubishi Tanabe, Sumitomo Dainippon, AstraZeneca, Ono, and Taisho pharm and has received research funding from Bristol-Myers Squibb and Gilead Sciences. Dr. Yoneda has received research funding from AbbVie, Bayer Pharma, EA pharma, Otsuka Pharmaceutical, and Sumitomo Pharmaceutical."

Reviewers' comments:

Reviewer's Responses to Questions

**Comments to the Author**

1. Is the manuscript technically sound, and do the data support the conclusions?

Reviewer #1: Partly

Reviewer #2: Yes

2. Has the statistical analysis been performed appropriately and rigorously? 

Reviewer #1: No

Reviewer #2: Yes

3. Have the authors made all data underlying the findings in their manuscript fully available?

Reviewer #1: Yes

Reviewer #2: Yes

4. Is the manuscript presented in an intelligible fashion and written in standard English?

Reviewer #1: No

Reviewer #2: Yes

5. Review Comments to the Author

Reviewer #1: In this study, Ito et al. evaluates Mac-2 binding protein glycosylation isomer and the FIB-4 index as biomarkers for NASH diagnosis. They find that a combination of these two factors can accurately predict NASH. Although a very interesting study, there are some concerns:

• Abstract: There are some grammatical issues.

o The word ‘of’ should be added between “10%” and “non-alcoholic fatty liver disease”.

o There needs to be a period at the end of the following sentence “Therefore, reliable noninvasive biomarkers for predicting NASH are required to prevent unnecessary liver biopsies”.

• Results:

o In figures 1a and b, there were no statistics show between the stages. The authors claim that M2BPGi levels and FIB-4 index can estimate the stage of fibrosis. Can every stage be differentiated using these values? To address this, the authors should show if the values are statistically different between stages.

o In figure 2, the authors should provide a possible explanation to why values were lower for scores 6 and 7 of NAS. Where they statistically lower? Showing statistics or stating the statistics would be helpful.

o The axes on the figures need to be properly labeled. For example, Figure 1 needs an x-axis label, and the y-axis should be labeled “M2BPGi (COI)”

• Discussion:

o The authors should address their recommendation for if a patient tests positive for one biomarker and not the other. Should the patient get a liver biopsy? Even though the two biomarkers are more accurate together, are they still accurate enough of their own?

Reviewer #2: In this study, the authors aimed to evaluate the performance of two non-invasive fibrosis markers, Mac-2 binding protein glycosylation isomer (M2BPGi), and the FIB-4 index for predicting the fibrosis staging, NAFLD activity scoring (NAS) index, and NASH. The authors demonstrated that both markers were well-correlated with each other in most cases, no correlation was found in some patients. Compared to the FIB-4 index or the M2BPGi alone, a combination of the two showed a higher specificity, PPV, and accuracy for NASH diagnosis. Overall, the findings from this study could have interesting implications. Specific points need to be considered are listed below.

1. As the authors mentioned that M2BPGi can be elevated due to causes other than liver diseases, it would be better to exclude patients with other diseases, such as breast cancer and other diseases listed in Table 3.

2. It would be interesting to check whether the M2BPGi is Wisteria floribunda agglutinin (WFA)-positive since previous findings suggest that WFA-positive M2BP levels predicted the presence of advanced disease and correlated strongly with fibrosis stage (PMID: 30161179).

3. The authors need to check the text carefully as there are many typos. For example, there’s abbreviation for HCC in Table 3, which does not even mention HCC in the table.

6. PLOS authors have the option to publish the peer review history of their article (what does this mean?). If published, this will include your full peer review and any attached files.

Reviewer #1: No

Reviewer #2: No

---

## [Author Response · Author response to Decision Letter 0]

19 Oct 2022

Reviewer #1

Comment: In this study, Ito et al. evaluates Mac-2 binding protein glycosylation isomer and the FIB-4 index as biomarkers for NASH diagnosis. They find that a combination of these two factors can accurately predict NASH. Although a very interesting study, there are some concerns:

Response:

Thank you so much for your thoughtful and meaningful comment.

Abstract 1:

o The word ‘of’ should be added between “10%” and “non-alcoholic fatty liver disease”.

Response:

A native English speaker has proofread the revised manuscript.

Abstract 2:

o There needs to be a period at the end of the following sentence “Therefore, reliable noninvasive biomarkers for predicting NASH are required to prevent unnecessary liver biopsies”.

Response:

Thank you for pointing out the missing period. We have added the period (line 29, page 2).

Result 1:

o In figures 1a and b, there were no statistics show between the stages. The authors claim that M2BPGi levels and FIB-4 index can estimate the stage of fibrosis. Can every stage be differentiated using these values? To address this, the authors should show if the values are statistically different between stages.

Response:

Thank you for your comment. We have statistically analyzed the association between fibrosis stage and M2BPGi and between fibrosis stage and FIB-4 index using Spearman’s rank correlation coefficient.

The correlation coefficient between the fibrosis stage and M2BPGi was 0.553, with a P value less than 0.0001. The correlation coefficient between the fibrosis stage and FIB-4 index was 0.577, with a P value less than 0.0001. We have added these results in Figures 1A and 1B. In addition, we have changed the expression in the statistical analyses section of the Methods to “Spearman’s rank correlation coefficient was used when necessary.” (Lines 138-139, page 7) We also added the results of Spearman’s rank correlation coefficient in the Results section of the main text (Lines 172-173, page 9).

Result 2:

o In figure 2, the authors should provide a possible explanation to why values were lower for scores 6 and 7 of NAS. Where they statistically lower? Showing statistics or stating the statistics would be helpful.

Response:

Thank you for your comment. We have statistically analyzed the association between NAS and M2BPGi and between NAS and FIB-4 index using Spearman’s rank correlation coefficient. The correlation coefficient between NAS and M2BPGi was 0.332, with a P value of 0.0012. The correlation coefficient between NAS and FIB-4 index was 0.333 with a P value less than 0.0011. We have added these results in Figures 2A and 2B and the results section of the main text (Lines 188-189, page 10). We have also added the following sentences in the Discussion section of the main text (Lines 252-257, page 14), “There was an overall significant difference in the association between the NAS score and the M2BPGi or FIB-4 index. While there was no significant difference between scores 6 and 7 and other scores, scores 6 and 7 tended to be low. Therefore, because steatosis and lobular inflammation decrease when fibrosis is severely progressed, fibrosis is considered more advanced in NAS scores of 4 and 5 than in NAS scores of 6 and 7.

Result 3:

o The axes on the figures need to be properly labeled. For example, Figure 1 needs an x-axis label, and the y-axis should be labeled “M2BPGi (COI)”

Response:

Thank you for your comment. We have corrected Figures 1 and 2.

Discussion:

o The authors should address their recommendation for if a patient tests positive for one biomarker and not the other. Should the patient get a liver biopsy? Even though the two biomarkers are more accurate together, are they still accurate enough of their own?

Response:

Thank you for your important comment. We have added the following sentences in the concluding part of the discussion (Lines 320-324, page 17), “When M2BPGi is high and the FIB-4 index is low, the presence of diabetes mellitus, rheumatic disease, arteriosclerosis, and malignancy should be examined. Furthermore, when the M2BPGi is low and the FIB-4 index is high, other causes of platelet decrease or AST increase other than liver fibrosis should be examined. If no cause other than liver fibrosis is found to cause the discrepancy between M2BPGi and FIB-4 index, liver biopsy should be considered again.”

Reviewer #2: In this study, the authors aimed to evaluate the performance of two non-invasive fibrosis markers, Mac-2 binding protein glycosylation isomer (M2BPGi), and the FIB-4 index for predicting the fibrosis staging, NAFLD activity scoring (NAS) index, and NASH. The authors demonstrated that both markers were well-correlated with each other in most cases, no correlation was found in some patients. Compared to the FIB-4 index or the M2BPGi alone, a combination of the two showed a higher specificity, PPV, and accuracy for NASH diagnosis. Overall, the findings from this study could have interesting implications. Specific points need to be considered are listed below.

Response:

Thank you so much for your thoughtful and meaningful comment.

Point 1. As the authors mentioned that M2BPGi can be elevated due to causes other than liver diseases, it would be better to exclude patients with other diseases, such as breast cancer and other diseases listed in Table 3.

Response:

Thank you for your important comment. We have added the following sentences in the concluding part of the discussion (Lines 320-324, page 17), “When M2BPGi is high and the FIB-4 index is low, the presence of diabetes mellitus, rheumatic disease, arteriosclerosis, and malignancy should be examined. Furthermore, when the M2BPGi is low and the FIB-4 index is high, other causes of platelet decrease or AST increase other than liver fibrosis should be examined. If no cause other than liver fibrosis is found to cause the discrepancy between M2BPGi and FIB-4 index, liver biopsy should be considered again.”

Point 2. It would be interesting to check whether the M2BPGi is Wisteria floribunda agglutinin (WFA)-positive since previous findings suggest that WFA-positive M2BP levels predicted the presence of advanced disease and correlated strongly with fibrosis stage (PMID: 30161179).

Response:

Thank you for your comment. M2BPGi is the same as the WFA-positive M2BP we have developed (Kuno A, Ito K et al. Sci Rep. 2013 PMID: 23323209). We have called it “WFA-positive M2BP” previously; however, currently, we call it “M2BPGi” in clinical settings.

Point 3.. The authors need to check the text carefully as there are many typos. For example, there’s abbreviation for HCC in Table 3, which does not even mention HCC in the table.

Response:

Thank you for pointing out our mistake. In table 3, we have deleted HCC and replaced it with “non-alcoholic steatohepatitis” for NASH below the table, which was our original intention. We have also added the definitions of NASH and M2BPGi below Table 2. Finally, we have also added the definition of M2BPGi in the Figure legends.

---

## [Decision Letter · Decision Letter 1]

26 Oct 2022

Mac-2 binding protein glycosylation isomer, the FIB-4 index, and a combination of the two as predictors of non-alcoholic steatohepatitis

PONE-D-22-23754R1

Dear Dr. Ito Kiyoaki,

We’re pleased to inform you that your manuscript has been judged scientifically suitable for publication and will be formally accepted for publication once it meets all outstanding technical requirements.

Kind regards,

Gianfranco D. Alpini

Academic Editor

PLOS ONE

Additional Editor Comments (optional):

Reviewers' comments:

Reviewer's Responses to Questions

**Comments to the Author**

1. If the authors have adequately addressed your comments raised in a previous round of review and you feel that this manuscript is now acceptable for publication, you may indicate that here to bypass the “Comments to the Author” section, enter your conflict of interest statement in the “Confidential to Editor” section, and submit your "Accept" recommendation.

Reviewer #1: All comments have been addressed

Reviewer #2: All comments have been addressed

2. Is the manuscript technically sound, and do the data support the conclusions?

Reviewer #1: Yes

Reviewer #2: (No Response)

3. Has the statistical analysis been performed appropriately and rigorously? 

Reviewer #1: Yes

Reviewer #2: (No Response)

4. Have the authors made all data underlying the findings in their manuscript fully available?

Reviewer #1: Yes

Reviewer #2: (No Response)

5. Is the manuscript presented in an intelligible fashion and written in standard English?

Reviewer #1: Yes

Reviewer #2: (No Response)

6. Review Comments to the Author

Reviewer #1: The authors addressed all the comments and made the necessary changes. It is a very interesting study and the revised manuscript is satisfactory to meet all the requirements. I have no additional comments for the authors.

Reviewer #2: (No Response)

7. PLOS authors have the option to publish the peer review history of their article (what does this mean?). If published, this will include your full peer review and any attached files.

Reviewer #1: No

Reviewer #2: No

---

## [Editor Report · Acceptance letter]

1 Nov 2022

PONE-D-22-23754R1 

Mac-2 binding protein glycosylation isomer, the FIB-4 index, and a combination of the two as predictors of non-alcoholic steatohepatitis 

Dear Dr. Kiyoaki:

I'm pleased to inform you that your manuscript has been deemed suitable for publication in PLOS ONE. Congratulations! Your manuscript is now with our production department. 

Kind regards, 

on behalf of

Dr. Gianfranco D. Alpini 

Academic Editor

PLOS ONE